# Optimisation of Planning Parameters for Machining Blade Electrode Micro-Fillet with Scallop Height Modelling

**DOI:** 10.3390/mi12030237

**Published:** 2021-02-26

**Authors:** Yue Liu, Zhanqiang Liu, Wentong Cai, Yukui Cai, Bing Wang, Guoying Li

**Affiliations:** 1School of Mechanical Engineering, Shandong University, Jinan 250061, China; sduliuyue@mail.sdu.edu.cn (Y.L.); efficientc@126.com (W.C.); caiyukui@sdu.edu.cn (Y.C.); sduwangbing@sdu.edu.cn (B.W.); sduliguoying@mail.sdu.edu.cn (G.L.); 2Shenzhen Research Institute of Shandong University, A301 Virtual University Park in South District of Shenzhen, Shenzhen 518000, China; 3Center for Experimental Mechanical Engineering Education, Key Laboratory of High Efficiency and Clean Mechanical Manufacture of MOE/Key National Demonstration, Shandong University, Jinan 250061, China

**Keywords:** scallop height model, cutting tool path planning, micro-fillet, profile roughness, cutting step, tool tilt angle

## Abstract

Aero-engine blades are manufactured by electroforming process with electrodes. The blade electrode is usually machined with five-axis micromilling to get required profile roughness. Tool path planning parameters, such as cutting step and tool tilt angle, have a significant effect on the profile roughness of the micro-fillet of blade electrode. In this paper, the scallop height model of blade electrode micro-fillet processed by ball-end milling cutter was proposed. Effects of cutting step and tool tilt angle the machined micro-fillet profile roughness were predicted with the proposed scallop height model. The cutting step and tool tilt angle were then optimised to ensure the contour precision of the micro-fillet shape requirement. Finally, the tool path planning was generated and the machining strategy was validated through milling experiments. It was also found that the profile roughness was deteriorated due to size effect when the cutting step decreased to a certain value.

## 1. Introduction

The machining precision and surface quality of the blade electrode have a significant influence on the machining quality of engine blades in the electrotyping process of aero-engine impeller blades. However, the machining of the micro-fillet on a blade electrode is challenging because of its high precision requirement, small size and considerable curvature variation. Five-axis micromilling technology is advantageous because it is suitable for processing three-dimensional (3D) microstructures [1,2,3]. The tool path parameters in machining programming, especially the tool tilt angle and cutting step, affect the scallop height and surface formation process, thus determining the profile roughness.

The scallop height model [4] proposed in 1995 can be used as a prediction model of surface roughness. Figure 1 displays the scallop height model with a ball-end milling cutter given a flat processing surface, which is the simplest case in milling.

The relationship between the horizontal cutting step of the tool path and the scallop height can be obtained by geometric calculation:(1)$$h=R-\sqrt{R^{2}-{\left(\frac{s}{2}\right)}^{2}}$$
where *h* is the scallop height. *R* is the radius for ball-end milling cutter, and *s* represents the horizontal cutting step of tool path. However, for complex freeform surfaces, the scallop height is inapplicable. On this basis, Hsi-Yung Feng et al. [5] optimised a constant scallop height (iso-scallop height) model of three-axis surface machining. Tournier C. et al. [6] proposed iso-scallop height tool paths to solve the problem of curvature discontinuities in surface machining. Min Cheng [7] considered the true 3D configuration in the iso-scallop height model. Shanshan Chen et al. [8] generated a tool path with uniform iso-scallop height considering the diverse curvature radii in the ultra-precision grinding of freeform surfaces.

In the scallop height model of flat surface machining, the scallop height h seems to be affected only by the tool radius and cutting step. However, in complex microstructures and in actual processing, the scallop height and surface roughness are also influenced by other factors. Thus, the influence of other parameters, such as tool run-out in the axial and radial directions [9], cutting depth and width [10] and feed rate [11], on surface roughness has been investigated based on the classical scallop height model. Experimental results show that the surface generation model can predict the surface roughness. Hendriko [12] employed an extended analytical boundary method to attain the scallop height in freeform machining considering the inclination angle. The method and model were verified accurately. Fangyu Peng et al. [13] investigated the influence of the cutter’s initial phase angle, path interval, feed rate per tooth and tool vibration on surface micro-topography by modelling the scallop height, which was verified effectively. Jenq-Shyong Chen et al. [14] studied the scallop height model in plane milling with a ball-end milling cutter and found that the influence of the feed-interval scallop height can be three to four times the path interval; in addition, they stated that the surface roughness reaches the allowable error range when the inclination angle is up to 10°.

The above studies reveal that the scallop height model can simulate surface generation and roughness and provide guidance for tool path planning and parameter selection. However, works on path planning based on scallop height are mostly aimed at theoretical studies or practical applications of simple structures. Therefore, the current paper established a scallop height model of the micro-fillet to predict its surface roughness. The influence of the cutting step (tool path interval) and tool tilt angle on the scallop height and surface roughness was analysed in the micro-fillet micromachining of a blade electrode. The tool path planning was generated. Through cutting experiments, a streamline machining strategy was selected to ensure contour accuracy. Last, the cutting step and tool tilt angle were optimised via model analysis and experiments. In this paper, the scallop height model of micro-fillet in micromilling was established innovatively. The scallop heights on different positions of micro-fillet were calculated. The influence of tool path planning parameters such as cutting step and tool tilt angle on scallop height were analysed. The cutting experiments were conducted. It was found that the scallop height was a critical factor for the surface roughness. Finally, the cutting step and tool tilt angle were optimally selected. The application of the scallop height model of micro-fillet for the blade electrode in the micro-milling provided a guiding significance and a successful case for the theoretical analysis to improve the surface quality in complex microstructure processing process.

## 2. Micro-Fillet Structure of Blade Electrode and Its Geometric Model

Aero-engine blades have the characteristics of complex shapes, large dimension spans and high precision. Machining deformation readily occurs in the numerical control machining blade, resulting in low machining accuracy. The electroforming process can not only easily obtain complex special structure parts, but also can accurately copy the surface profile and microstructure, achieve high dimensional accuracy and excellent surface quality. However, electroforming will also bring the scars on the core mould to the product, so when electroforming the blade, the size precision and surface quality of the blade electrode have very high requirements. Figure 2 shows the assembly structure of the blade electrode. The blade shape is between the convex of blade electrode and the concave of blade electrode. The machining of the micro-fillet on the blade electrode is a difficult point.

The electrode of a concave surface is shown in Figure 3a. The minimum fillet radius of the profile of the convex and concave surfaces is 0.2 mm. The micro-fillet of a blade electrode is illustrated in Figure 3b. As can be seen from Figure 3, the curvature of the electrode surface changes greatly, and the concave and convex surfaces are connected. The micro-fillet is small and extends along the free curves. Stress can easily concentrate during processing, especially in the micro-fillet, and it is difficult for traditional triaxial machining to satisfy both contour accuracy and profile roughness requirements.

To study the influence of tool path planning parameters on contour accuracy and profile roughness of the electrode microstructure, which is the hardest machining part, we simplified the electrode model by retaining only the micro-fillet shape. The radius of the micro-fillet of the micro-fillet of the convex and concave surface is 0.2 mm. The micro-fillet extends along the free curves in the simplified model. The simplified model and its micro-fillet shape, which will be subjected to practical processing experiments, are illustrated in Figure 4.

## 3. Scallop Height Calculation of Micro-Fillet

To finish the machining processes, an appropriate tool tilt angle and horizontal cutting step of the cutting tool path must be selected to ensure processing efficiency and improve profile roughness. The surface scallop height can be used to predict the profile roughness and provide the basis for tool path generation and parameter selection.

First, a distinction is made between the B-axis deflection angle *α* and tool tilt angle *κ*. According to International Standardization Organization (ISO) regulations, the right-handed rectangular coordinate system is used when describing the motion of Computer Numerical Control (CNC) machine tools; the coordinate axis parallel to the main axis is defined as the z axis, and the rotation coordinates around the x, y and z axes are A, B and C, respectively [15]. Therefore, the B-axis deflection angle is the deflection angle of the tool spindle along the B-axis direction in the machine tool coordinate system [16]. The B-axis deflection angles *α* of plane milling and bevel milling are represented in Figure 5a,b, respectively. As seen in Figure 5, the B-axis deflection angle is related to the machine coordinate system but not to the surface curvature of the workpiece.

The tool tilt angle *κ* is the inclination angle of the tool in the radial feed direction *c*, based on the normal *N* to the surface of the workpiece [17]. In Figure 5, *f* is the feed direction, *c* is the radial feed direction, *N* is the normal of the machining point, and *f c N* conforms to the right-hand rule. The positive or negative value of the tool tilt angle *κ* is determined according to this rule. When *κ* > 0, the tool tilts toward the radial feed direction *c*; when *κ* < 0, the tool tilts toward the direction opposite *c*. This angle is changed according to the curvature.

According to the original scallop height model of flat surface machining, the scallop height model of the micro-fillet processed by the ball-end milling cutter in this study is established via the geometric method, as illustrated in Figure 6. Points *O*, *O′* and *O″* form the centre of the micro-fillet and the ball-end milling cutter at the initial and final positions.

When the B-axis deflection angle *α* is fixed, the tool tilt angle *κ* at each cutting point varies with the cutting process. The geometric relationships on the cutting point between the cutter and the micro-fillet vary with the cutting position.

These positions can be summarised into three situations represented as three regions. (1) Region I: The tool tilt angle *κ* is negative. (2) Region II: The tool nose cuts the fillet, that is, the centre *O* of the fillet is on the cutter axis (*κ* = 0). (3) Region III: The tool tilt angle *κ* is positive. The intermediate variable *θ* is introduced to calculate the scallop height *h*. The relationship between *h* and *θ* is described in region I.

In Figure 6, *s* is the horizontal cutting step, and *R* and *r* are the radii of the micro-fillet and the ball-end milling cutter, respectively. First, *θ* is obtained by *κ*, and the scallop height *h* is calculated based on geometric derivation. In regions I–III, the geometric relationship and expression of *κ* and *θ* are constantly changing, while the geometric relationship between *θ* and *h* is settled. Therefore, only the geometric relationship between *κ* and *θ* is enlarged in regions II and III in Figure 6 to clearly calculate the result. In region I, where *κ* < 0, *θ* can be solved and expressed as
(2)$$\begin{array}{l} \theta =\frac{180{}^{\circ}-\left(90{}^{\circ}-\kappa \right)-\arccos\left[\frac{s\cos\alpha -\left(R-r\right)\sin\kappa }{R-r}\right]}{2}\\ =\frac{90{}^{\circ}+\kappa -\arccos\left[\frac{s\cos\alpha -\left(R-r\right)\sin\kappa }{R-r}\right]}{2} \end{array}$$

In region II, where the tool nose cuts the fillet (*κ* = 0),
(3)$$\theta =\frac{90{}^{\circ}-\arccos\left(\frac{s\cos\alpha }{R-r}\right)}{2}$$

In the period after the tool nose cuts the fillet (*κ* > 0), *θ* can be solved and expressed as
(4)$$\theta =\frac{90{}^{\circ}-\kappa -\arccos\left[\frac{s\cos\alpha +\left(R-r\right)\sin\kappa }{R-r}\right]}{2}$$

In consequence, the scallop height *h* can be represented by *θ* as
(5)$$h=R-\left[\left(R-r\right)\cos\theta +\sqrt{r^{2}-{\left(R-r\right)}^{2}\sin^{2}\theta }\right]$$

The specific data can be substituted into Equations (2)–(5) to calculate the scallop height value at each position, where *R* = 0.2 mm, *r* = 0.15 mm and *α* = 15°. The cutting step *s* is set as 0.005 mm, 0.015 mm, 0.025 mm, 0.035 mm and 0.045 mm to obtain the scallop height under each cutting step. The tool tilt angle is changed by the cutting position. Thus, the equation for calculating the tool tilt angle *κ* and the central angle *φ* varying from 0° to 90° is as follows:(6)φ=90°−α−κ; κ < 0,
(7)φ=90°−α; κ = 0,
(8)φ=90°−α+κ; κ > 0.

Therefore, the scallop height at each position of the micro-fillet, corresponding to the change in the central angle from 0° to 90°, is calculated. The scallop height at each position of the micro-fillet under 0.005–0.045 mm cutting step is shown in Figure 7. The curve of the central angle and the scallop height of the micro-fillet is drawn in Figure 7a. Correspondingly, the theoretical prediction of the scallop height curve on the micro-fillet can be obtained by fitting the curve to the micro-fillet, as shown in Figure 7b. As indicated by the combination of Equations (6)–(8), when the central angle *φ* changes from 0° to 90°, the tool tilt angle *κ* changes from negative to positive. When the tool tilt angle *κ* is near 0°, the scallop height is relatively small.

From the fillet scallop height model, three main conclusions are determined: (1) The scallop height in the axial direction increases with the cutting step. (2) At the same cutting step, the scallop height on the micro-fillet decreases and then increases along the tangential direction. When the central angle is 25°–70°, the scallop height is smaller. This change is due to the different tilt angles at the various positions. The difference between the maximum and minimum scallop heights at the various positions of the micro-fillet increases with the cutting step. For example, when the cutting step is 0.005 mm, the difference between the maximum and minimum scallop heights is 0.46 μm; when the cutting step is 0.045 mm, the difference is 8.18 μm. (3) The fixed tool deflection angle in the B axis, which corresponds to the varying tool tilt angle at the different positions, affects the position of the minimum scallop height. The area with the gentle surface morphology and small scallop height can be selected in combination with the experiment to obtain the tool tilt angle range.

## 4. Experiments

### 4.1. Tool Path Planning

The software Hyper MILL 2018.1(Open mind Company, München, Germany) is used to generate different tool paths for the micro-fillet machining. The micro-fillet shapes are processed by 3D and 5X isometric finishing after rough machining. First, the machining strategy in the process of five-axis machining is optimised to ensure the contour precision of the fillet shape microstructure. Then, the influence of the cutting step and tool tilt angle during five-axis machining on the profile roughness of the fillet shape microstructure is investigated under the optimised machining strategy.

#### 4.1.1. Experiment Design of Machining Strategy

To ensure contour accuracy, a machining strategy which determines the feeding shape and the way of feeding and relieving is optimised through experiments. This strategy is then divided into two types, namely, streamline and isometric, which correspond to direction-parallel and contour-parallel tool path patterns. The streamline machining strategy makes the cutting tool move along the streamline according to the shape trend. The isometric machining strategy makes the cutting tool move along the circumference. The two types of machining strategy and the tool paths of the micro-fillet in 5X isometric finishing are demonstrated in Figure 8.

The two machining strategies—streamline and isometric—are adopted separately in the following experiments while the parameters, such as the cutting step and tool B-axis deflection angle, are kept constant. The profile roughness and contour accuracy of the processed samples are then measured.

#### 4.1.2. Experiment Design of Cutting Step and Tool B-Axis Deflection Angle

The cutting step is set as 0.005 mm, 0.015 mm, 0.025 mm, 0.035 mm and 0.045 mm to ensure contour accuracy and thus improve profile roughness.

In the experiment, the B-axis deflection angle of the cutting tool is constant and set as 15°. The influence of the tool tilt angle on the profile roughness of different parts of the micro-fillet is studied. The geometric position of the B-axis deflection angle of the cutting tool is shown in Figure 9.

### 4.2. Machining Experiments

Experiment device. The experimental equipment is the five-axis high-speed micromachining centre Kern 2522 with a Heidenhain iTNC 530 CNC system (Heidenhain Company, Traunreut, Germany). The maximum speed of the spindle of the machine is 50,000 rpm, and the machining accuracy of the machine for each workpiece can reach ±2.5 µm. The main function of the machine is precision machining of micro-parts and micromachining characteristics. The positions of the machine tool and the workpiece in the machining process are shown in Figure 10.

Material of blade electrode: 1Cr18Ni9Ti (Huahu Company, Shanghai, China) stainless steel was selected as the material of blade electrode. 1Cr18Ni9Ti is stainless steel (SUS321), and its micro-structure is Austenite. The numbers 1, 18 and 9 in 1Cr18Ni9Ti represent the content of carbon (‰), chromium (%) and nickel (%) respectively. The chemical compositions of 1Cr18Ni9Ti stainless steel are shown in Table 1. The mechanical properties of 1Cr18Ni9Ti stainless steel are shown in Table 2.

Cutting tool: High-speed steel, coated high-speed steel and coated cemented carbide tools are often used as cutting materials in milling stainless steel parts. In this research, an MSB230G2 MUGEN-COATING (NS Tool Company, Tokyo, Japan) high-precision profit ball-end mill, which has a ball-nose radius of 0.15 mm and precision of ±2 µm, is selected. The ball-end milling cutter used for finishing is shown in Figure 11.

Machining process. The processing technology of the simplified micro-fillet model is divided into rough machining, semi-finishing (3D isometric finishing) and finishing (5X isometric finishing). The detailed process parameters are shown in Table 3, where the horizontal and vertical cutting steps correspond to the cutting width and depth, respectively.

## 5. Results and Discussion

### 5.1. Optimisation of Machining Strategies for Contour Precision

For the study of the two machining strategies, as described in Section 4.1.1 (experimental design), micro-fillet samples were produced using the two strategies through experiments. The profile roughness and contour accuracy of the samples were observed with laser scanning confocal microscope (LSCM) Keyence VK-X250 (KEYENCE CO., LTD. Osaka, Japan). The LSCM is shown as Figure 12. Figure 13a shows a machined micro-fillet sample. The two-dimensional (2D) and 3D image of the micro-fillet are illustrated in Figure 13b,c, respectively.

The surface quality was evaluated by measuring the multilinear roughness (Ra) along the micro-fillet, as shown in Figure 14a,b. Five parts at the same position of the micro-fillet samples with different machining strategies were selected. Three sets of multilinear roughness were measured at the same position. The measurement data were summarised, and the multilinear roughness measurement results for the two machining strategies are shown in Figure 14c. The multilinear roughness values of the micro-fillet samples in the streamline and isometric machining strategies are Ra 1.86 µm and Ra 1.74 µm, respectively. Thus, the machining strategies have minor effects on profile roughness.

Five parts at the same position of the micro-fillet samples from the two machining strategies were selected; three sets of contour accuracy were measured at each position. As the radius of the micro-fillet was expected to be 0.2 mm, the contour of each section of the 3D image was obtained to fit the whole circle, as shown in Figure 15a. Then, the standard deviation between the ideal radius and the radius of the whole fitted circle was calculated, and the result was regarded as the evaluation standard of the contour accuracy. The smaller the standard deviation, the better the contour accuracy. The contour accuracy measurement results are collated in Figure 15b.

The contour precision values of the micro-fillet samples produced with the streamline and isometric machining strategies are 5.16 µm and 6.63 µm, respectively. The contour precision of the streamline machining strategy is evidently better than that of the isometric strategy. The concave shape of the micro-fillet has a height orientation and extends along the freeform curves. The streamline machining strategy feeds and cuts layer-by-layer along the height and shape of the micro-fillet during machining. By contrast, the isometric machining strategy is annular, that is, it cannot better guarantee the shape of the micro-fillet.

The experimental results showed that the profile roughness deviation between the streamline and isometric machining strategies is less than 0.12 μm, and the contour of the streamline machining strategy is more precise. Therefore, the streamline machining strategy is preferred for ensuring the contour precision of a micro-fillet.

### 5.2. Effect of Cutting Step on Profile Roughness

The streamline machining strategy was selected to ensure contour precision and profile roughness. The micro-fillet samples produced with cutting steps of 0.005 mm, 0.015 mm, 0.025 mm, 0.035 mm and 0.045 mm were processed on the five-axis machining centre to explore the effect of the cutting step on the profile roughness of a micro-fillet. Both 2D and 3D images of each micro-fillet can be obtained by LSCM. Five same positions at each micro-fillet sample were selected; three sets of multilinear roughness were measured and averaged at each position. The average value of the selected roughness samples was calculated. The measured multilinear roughness (Ra) data for each cutting step are shown in Table 4.

The multilinear roughness (Ra) results for each cutting step are demonstrated in Figure 16.

At the cutting step of 0.025 mm, the average multilinear roughness is about 1.56 µm. When the cutting step is greater than 0.025 mm, the multilinear roughness gradually increases because a larger cutting step results in a higher scallop height during the surface formation. This rule can be verified by the scallop height model of the micro-fillet. From the theoretical prediction, the scallop height can be expressed quantitatively by the cutting step without considering other factors. Based on the surface formation process, the smaller the cutting step, the lower the scallop height, which theoretically reduces the profile roughness. However, when the cutting step is less than 0.025 mm, the cutting step decreases while the profile roughness increases; this is due to the size effect of micromilling, which is neglected in the theoretical calculation [18,19]. In accordance with the minimum chip thickness model [20], when the cutting step is 0.005 mm and 0.015 mm smaller than the minimum chip thickness, the tool exerts extrusion pressure and plough force on the workpiece, resulting in serious surface deformation; in addition, the size effect mechanism is the main surface influence mechanism.

### 5.3. Effect of Tool Tilt Angle on Profile Roughness

According to the theoretical scallop height model, without the influence of the size effect, the scallop height and profile roughness at different positions of the micro-fillet are different. The processed surface contour data were collected and extracted to verify the accuracy of the scallop height model. The cutting step was fixed to study the influence of the tool tilt angle on the profile roughness. The five contour sets of the experimental micro-fillet at the cutting step of 0.025 mm are shown in Figure 17a. The theoretical prediction of the scallop height at the 0.025 mm cutting step is compared in Figure 17b,c.

According to the contour of the experimental micro-fillet and the theoretical prediction of the scallop height, the contours can be divided into three parts. The experimental results indicate that the roughness of parts A and C is smoother than that of part B. In the theoretical prediction scallop height model, the scallop height of B is the lowest, which contradicts the experimental results. Theoretically speaking, the scallop height on the surface of part B is small because the tool tilt angle is the smallest in this part. When the ball-end milling cutter cuts from top to bottom from the micro-fillet, the tool tilt angle decreases to 0 and then increases. Therefore, according to Equation (5), the scallop height of parts A and C is large.

In the experimental process, the spindle turns around the axis of the ball-end milling cutter, and the cutting speed increases gradually from the centre to the edge of the ball head. In the cutting process, the centre of the tool ball head cuts part B, where the cutting speed is the lowest and the cutting force is the maximum, resulting in worse surface roughness [21,22]. For parts A and C, the side edge of the tool ball head, which has a high cutting speed, participates in the cutting and causes a smooth profile roughness.

Therefore, the tool tilt angle not only influences the surface formation process through the scallop height model but also results in different cutting forces due to the various cutting speeds at the different cutting edges, thereby affecting the profile roughness of the micro-fillet. As the tool B-axis deflection angle changes dynamically during five-axis machining according to the curvature and inclination of the cutting surface, the tool tilt angle *κ* was obtained instead. According to the experimental results, the tool tilt angle *κ* corresponding to the region with the minimum profile roughness, that is, the region near part A should be selected. All the roughness contour samples measured at the cutting step of 0.025 mm were counted. The smooth part of the surface was substituted into the scallop height model established previously, and the tool tilt angle *κ’* was calculated to range from 38.88° to 62.02°.

Based on the experimental results, the above analysis reveals the co-influence of the size effect, surface formation mechanism and cutting force on the cutting position of the tool on the profile roughness of the micro-fillet. The minimum chip thickness is approximately 0.025 mm. When the cutting step is less than 0.025 mm, the size effect affects the profile roughness. When the cutting step is more than 0.025 mm, the effect of the cutting step on the profile roughness can be predicted by the scallop height model. The influence mechanism of tool deflection on the profile roughness of a micro-fillet mainly involves the scallop height model and the cutting force.

## 6. Conclusions

Tool path planning was generated to improve the profile roughness of the micro-fillet of aero-engine blades. A scallop height model was established for a micro-fillet processed by a ball-end milling cutter to predict the processed profile roughness through the geometric method. An experiment was performed to verify the scallop height model of the micro-fillet. First, the streamline machining strategy was preferred to ensure the contour accuracy of the micro-fillet. Second, the effect of the cutting step on the profile roughness was further studied. For optimisation of the cutting step, the machining experiment was designed to obtain a profile roughness under a 0.005–0.045 mm cutting step. Finally, the roughness contour curves were measured to analyse the tool tilt angle of the micro-fillet. Through the establishment of the model and experimental analysis, the following conclusions can be drawn.
From the scallop height model, (i) the scallop height of the surface increases with the cutting step. (ii) At the same cutting step, the scallop height on the micro-fillet decreases and then increases along the tangential direction. (iii) When the tool deflection angle in the B axis is fixed, the tool tilt angle changes from *κ* < 0 to *κ* > 0, thereby affecting the minimum position of the scallop height.The effect mechanism of the cutting step on the profile roughness includes the scallop height model and the size effect mechanism. The profile roughness is the smallest when the cutting step is 0.025 mm.Besides the scallop height model, the cutting force is an influence factor of the tool tilt angle on the profile roughness. The closer the ball-end mill to the centre of the cutter, the smaller the absolute value of the tool tilt angle, the greater the cutting force and the worse the profile roughness. During machining, the deflection angle of the B axis should be changed to guarantee that the tool tilt angle is within 38.88° to 62.02°, thus ensuring profile roughness.

## Figures and Tables

**Figure 1 micromachines-12-00237-f001:**
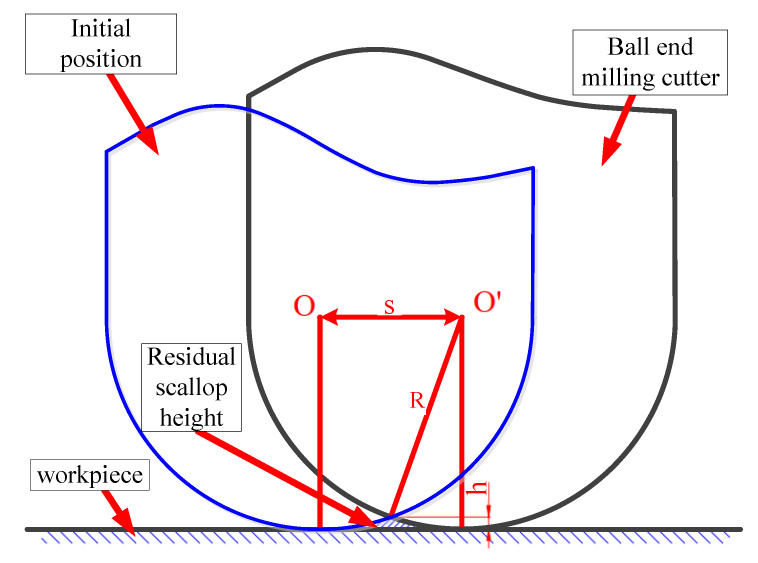
Scallop height in flat surface machining.

**Figure 2 micromachines-12-00237-f002:**
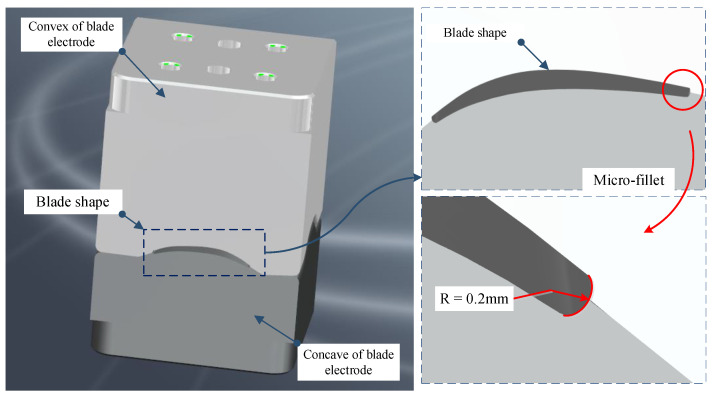
Assembly structure of blade electrode.

**Figure 3 micromachines-12-00237-f003:**
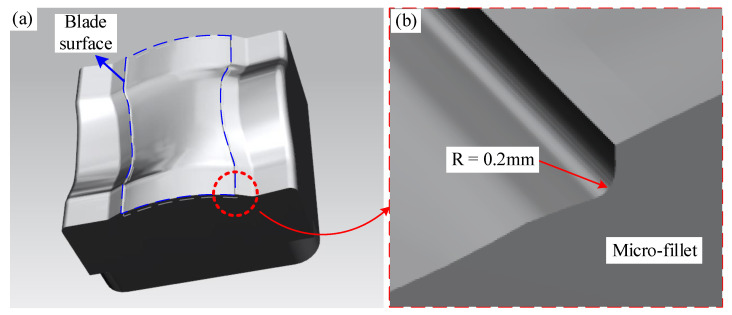
(**a**) Electrode of concave surface and (**b**) micro-fillet of blade electrode.

**Figure 4 micromachines-12-00237-f004:**
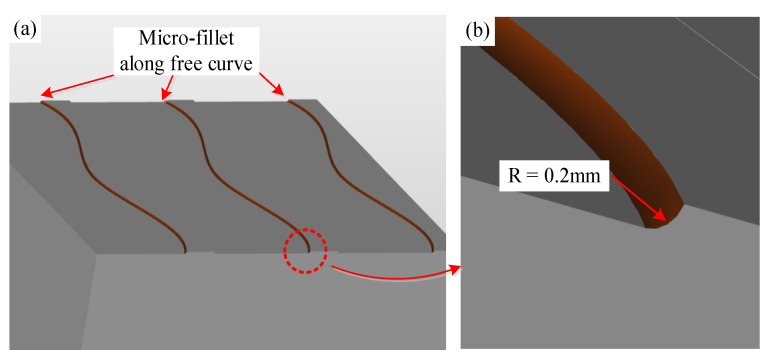
Simplified micro-fillet shape.

**Figure 5 micromachines-12-00237-f005:**
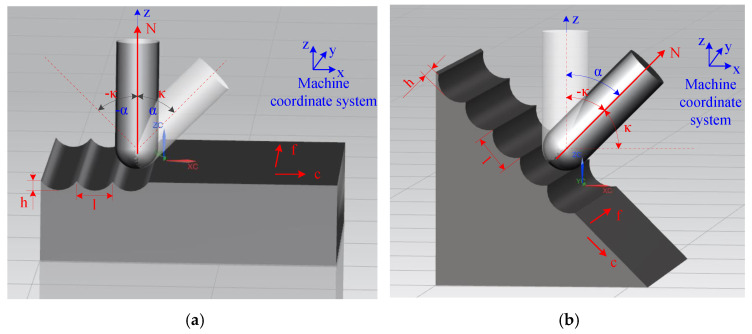
B-axis deflection angle *α* and tool tilt angle *κ* in (**a**) plane milling and (**b**) bevel milling.

**Figure 6 micromachines-12-00237-f006:**
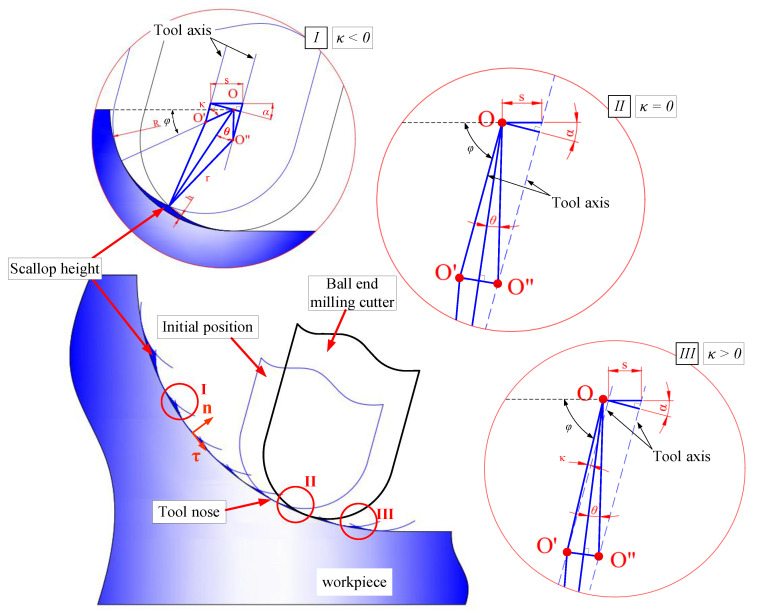
Scallop heights at different locations for micro-fillet machining.

**Figure 7 micromachines-12-00237-f007:**
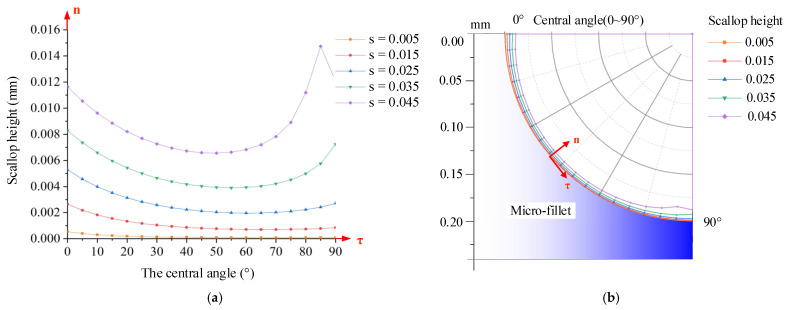
Scallop height at each position of micro-fillet under 0.005–0.045 mm cutting step: (**a**) curve of central angle and scallop height of micro-fillet, and (**b**) theoretical prediction of scallop height curve on micro-fillet.

**Figure 8 micromachines-12-00237-f008:**
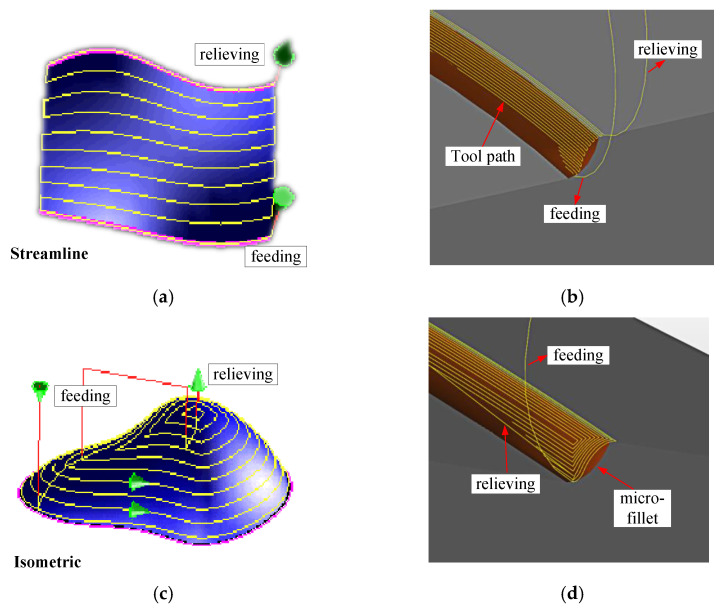
(**a**) Streamline machining strategy, (**b**) tool path of streamline machining strategy, (**c**) isometric machining strategy and (**d**) tool path of isometric machining strategy.

**Figure 9 micromachines-12-00237-f009:**
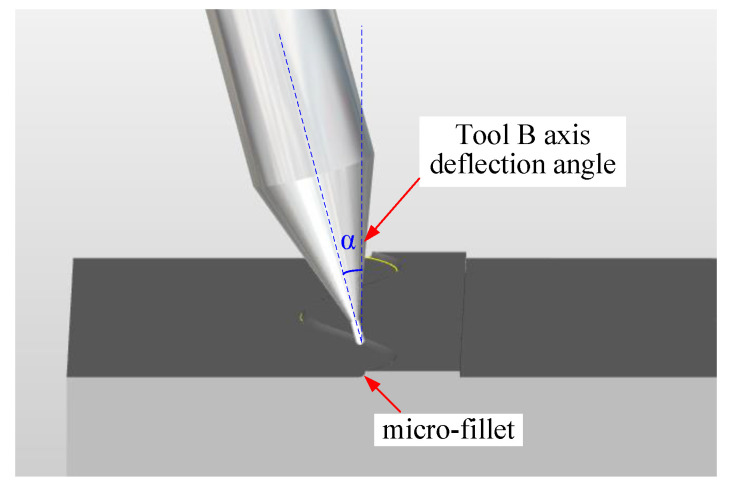
Tool path of tool B-axis deflection angle.

**Figure 10 micromachines-12-00237-f010:**
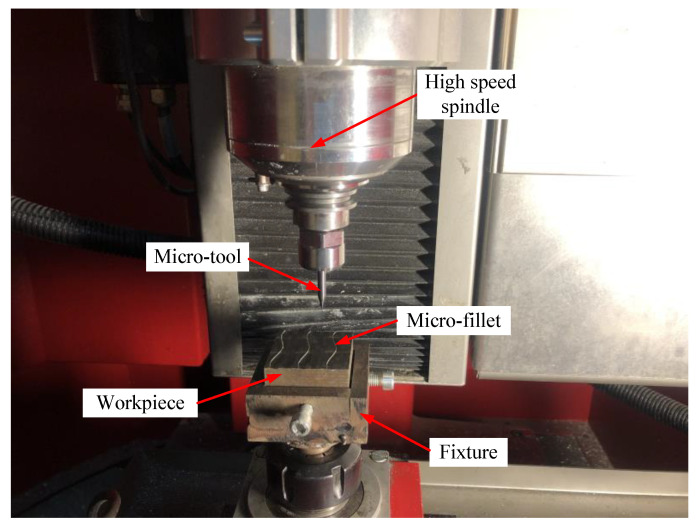
Experiment setup for micro-fillet machining.

**Figure 11 micromachines-12-00237-f011:**
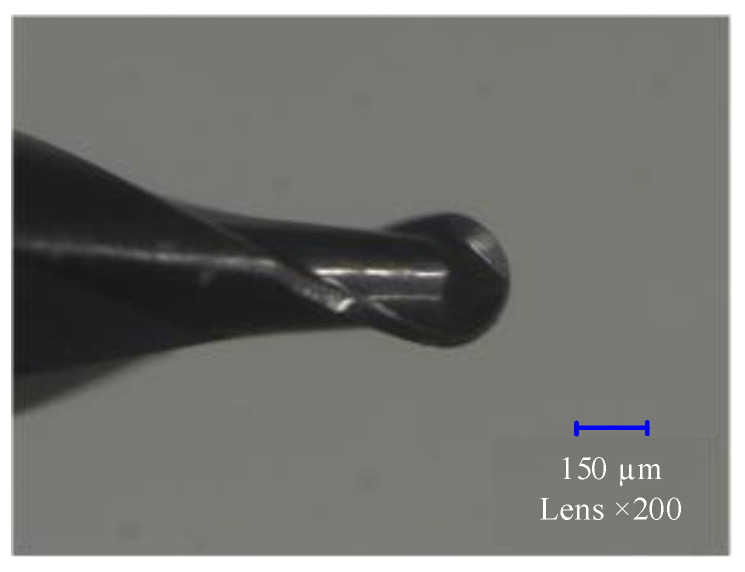
Micro ball-end milling cutter.

**Figure 12 micromachines-12-00237-f012:**
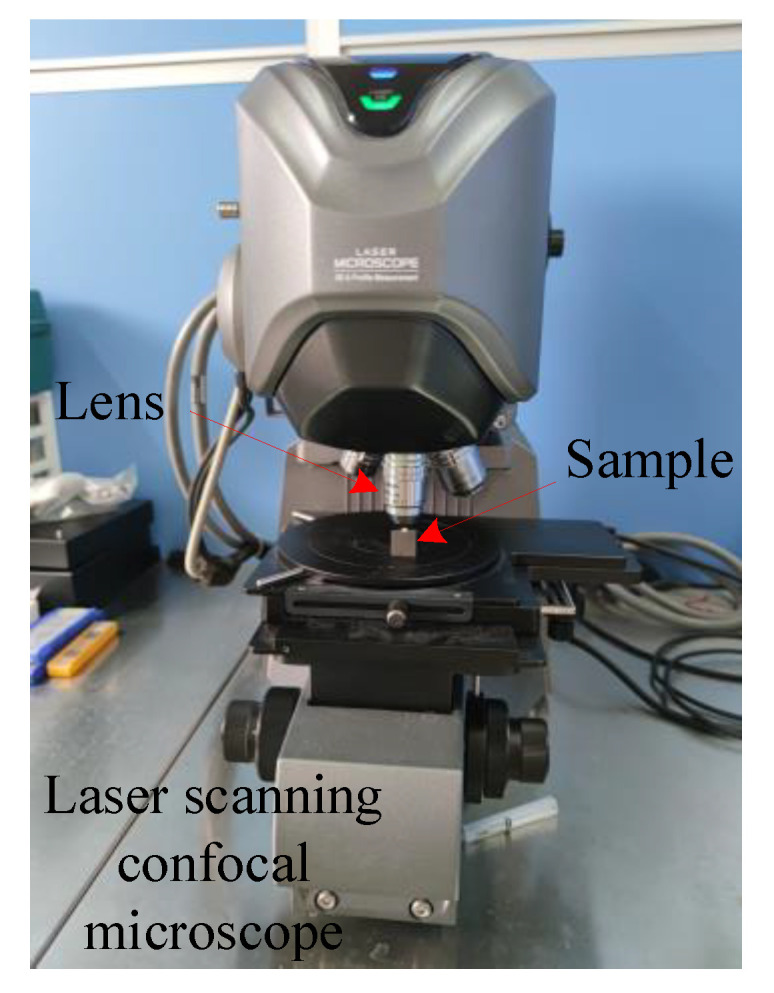
Laser scanning confocal microscope Keyence VK-X250.

**Figure 13 micromachines-12-00237-f013:**
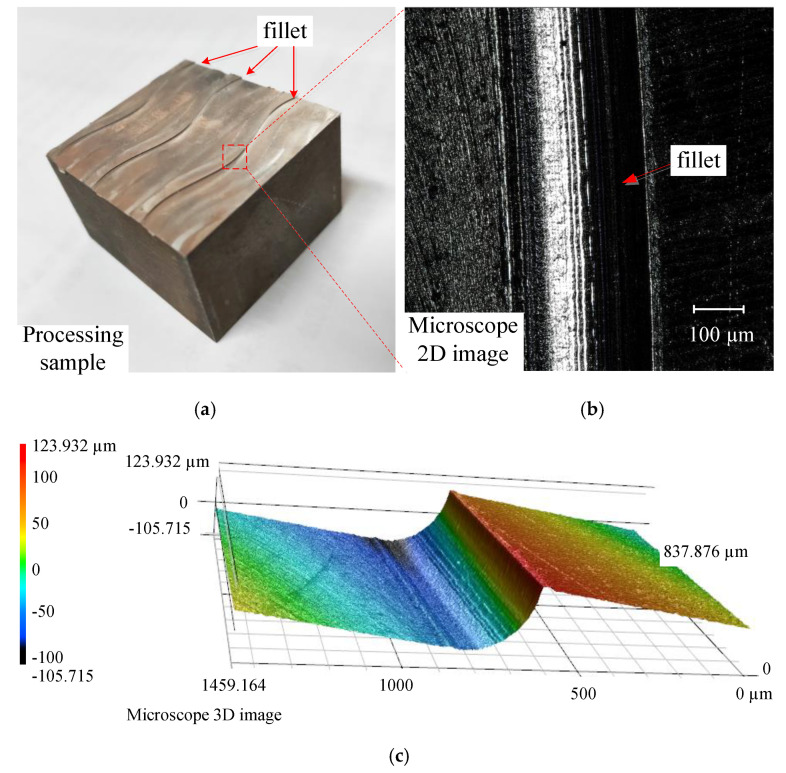
(**a**) Machining sample of fillet shape, (**b**) microscope 2D image, and (**c**) 3D image.

**Figure 14 micromachines-12-00237-f014:**
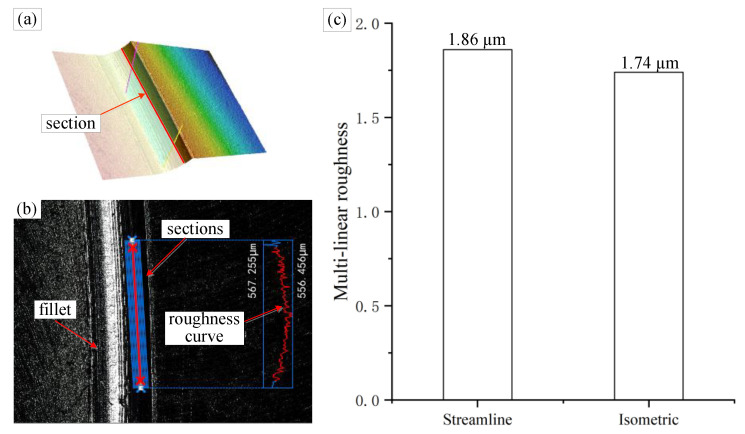
(**a**) Micro-fillet 3D image, (**b**) multilinear roughness measurement and (**c**) multilinear roughness result.

**Figure 15 micromachines-12-00237-f015:**
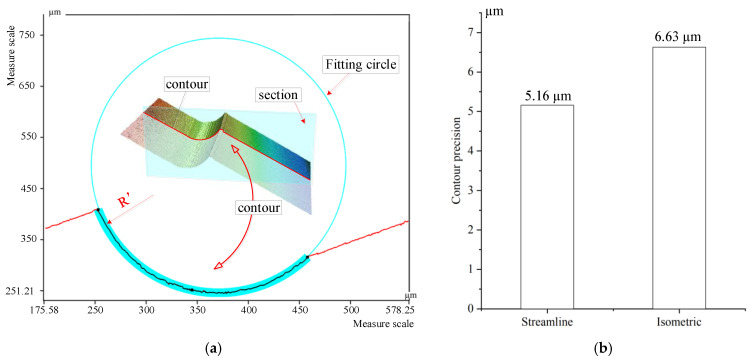
(**a**) Contour accuracy measurement. (**b**) Contour accuracy result.

**Figure 16 micromachines-12-00237-f016:**
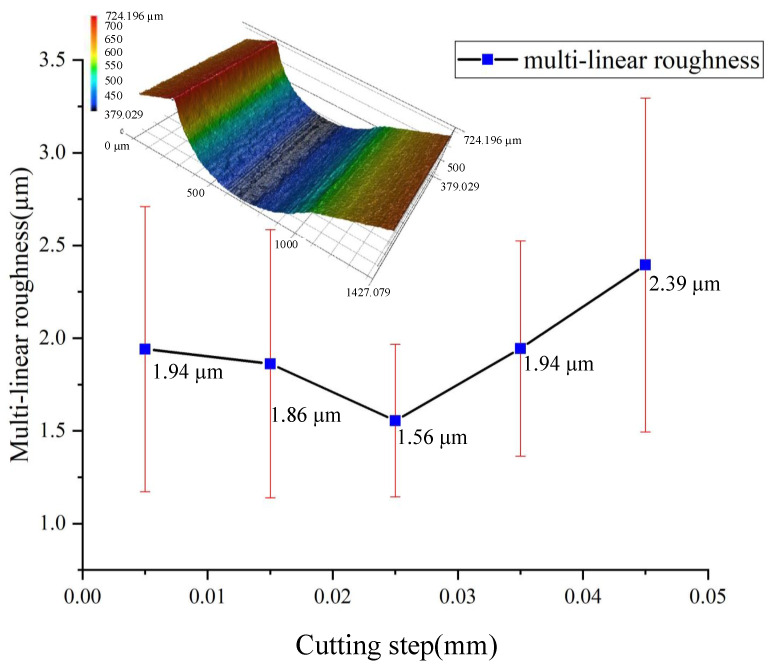
Multilinear roughness (Ra) for each cutting step.

**Figure 17 micromachines-12-00237-f017:**
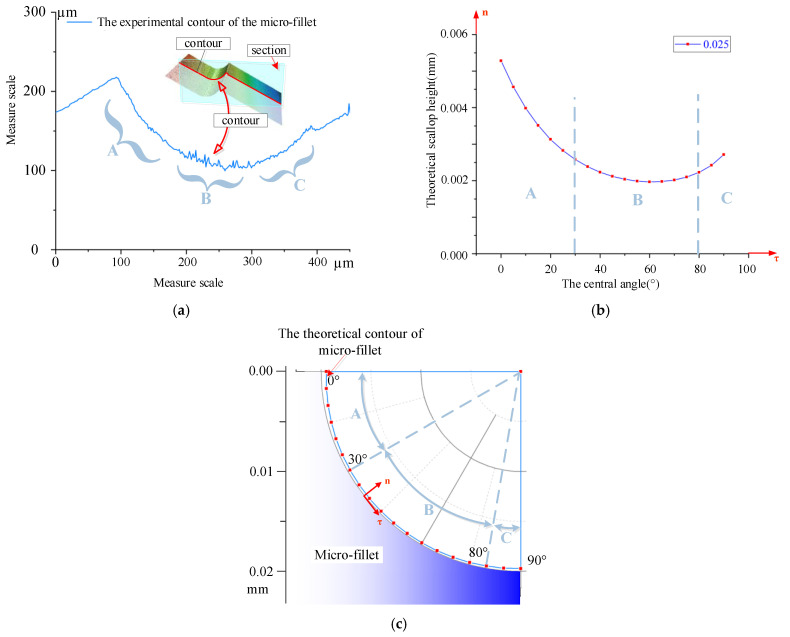
(**a**) Experimental contour of micro-fillet, (**b**) theoretical prediction of scallop height at 0.025 mm cutting step, and (**c**) theoretical prediction of scallop height on micro-fillet at 0.025 mm cutting step.

**Table 1 micromachines-12-00237-t001:** Chemical compositions of 1Cr18Ni9Ti stainless steel.

Chemical Composition	C	Si	Mn	S	P	Cr	Ni	Ti
**Value**	≤0.12%	≤1.00%	≤2.00%	≤0.030%	≤0.035%	17.00~19.00%	8.00~11.00%	5 (C%–0.02)~0.80%

**Table 2 micromachines-12-00237-t002:** Mechanical properties of 1Cr18Ni9Ti stainless steel.

Mechanical Properties	Tensile Strength	Yield Strength	Elongation	Necking-Down Rate	Hardness
**Value**	≥550 MN/m^2^	≥200 MN/m^2^	≥40%	≥50%	≤187 HB; ≤90 HRB; ≤200 HV

**Table 3 micromachines-12-00237-t003:** Machining process and parameters of micro-fillet.

Machining Parameters	Rough Machining	3D Isometric Finishing	5X Isometric Finishing
Milling Cutter Diameter (mm)	3	1	0.3
Spindle Speed (r/min)	8000	15,000	20,000
Feed Rate (mm/min)	800	400	50
Cutting Step (mm)	0.2	0.05	0.005, 0.015, 0.025, 0.035, 0.045
Allowance (mm)	0.05	0	0

**Table 4 micromachines-12-00237-t004:** Measured multilinear roughness (Ra) for each cutting step.

Cutting Step (mm)	Measured Position	Multilinear Roughness (μm)	Average (μm)	Cutting Step (mm)	Measured Position	Multilinear Roughness (μm)	Average (μm)
0.005	Part 1	2.90	1.94	0.015	Part 1	2.52	1.86
		1.26				1.87	
		0.91				1.22	
	Part 2	2.39			Part 2	2.17	
		2.30				1.85	
		1.50				1.07	
	Part 3	3.20			Part 3	2.94	
		3.16				1.72	
		1.95				1.06	
	Part 4	2.20			Part 4	2.90	
		1.56				1.92	
		1.02				1.21	
	Part 5	2.20			Part 5	2.55	
		1.61				1.71	
		0.95				1.19	
0.025	Part 1	1.83	1.56	0.035	Part 1	2.32	1.94
		1.74				2.08	
		1.17				1.67	
	Part 2	1.86			Part 2	2.77	
		1.26				2.45	
		0.90				1.61	
	Part 3	1.79			Part 3	2.87	
		1.41				2.60	
		1.09				2.07	
	Part 4	2.27			Part 4	1.94	
		1.77				1.29	
		1.31				0.88	
	Part 5	1.98			Part 5	1.68	
		1.71				1.62	
		1.26				1.32	
0.045	Part 1	2.81	2.39				
		2.72					
		1.50					
	Part 2	1.79					
		1.06					
		0.96					
	Part 3	2.90					
		2.86					
		2.77					
	Part 4	4.52					
		2.75					
		1.83					
	Part 5	2.87					
		2.68					
		1.94					

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
