# Peer review of "Optimisation of Planning Parameters for Machining Blade Electrode Micro-Fillet with Scallop Height Modelling"

_micromachines, 2021, doi:10.3390/mi12030237_

Round 1

Reviewer 1 Report

  • R and s are not in equation described.
  • Author should add description of novelty of his manuscript to the end of Introduction.
  • Fig.7 is not good readable. Font of text should be bigger.
  • Scale in Fig. 11 is not good readable.
  • In Figs. 15 and 16 is small font of text and is not readable.

Author Response

Thank you for your comments. The responses are in the attachment, please check it.

Reviewer 2 Report

The review concerns the work entitled "Optimisation of planning parameters for machining blade electrode micro-fillet with scallop height modelling".

This paper is not ready to publish - it requires some improvements and completed to be made according to the suggestions enlisted below:

  1. The keywords includ "surface topography", which, in my opinion,  had not been studied.
  2. I do not see the surface topography view and analysis of surface roughness (texture) parameters as Sa, Sq, Sz, Sp, Sv, Ssk, Sku etc.
    The Authors have been written only about Ra parametr, which is profile roughness parameter.
  3. There is no detail description of the research devices for surface roughness measurements. No information about measurement method, measurement area, amout of profile measurement, etc.
  4. There is no detail about machined material.
  5. Moreover, there are no descriptions of axis or scal (images) – fiugure 12, figure 14, figure16.
  6. Figure 15. How many profiles have been measured? What was the standard deviatioin? The table with data is missed.

Author Response

(The authors gave the same response as above.)

Reviewer 3 Report

Good job. A very interesting article.

Author Response

Thank you for your comments. 

Round 2

Reviewer 1 Report

Author reflected all my comments and improved his manuscript.

Reviewer 2 Report

My comments were included by Authors in the revised version of the manuscript.